# The effect of $(NH_4)_2SO_4$ on the freezing properties of non-mineral dust ice nucleating substances of atmospheric relevance

Soleil E. Worthy[1], Anand Kumar[1], Yu Xi[1], Jingwei Yun[1], Jessie Chen[1], Cuishan Xu[1], Victoria E. Irish[1], Pierre Amato[2], Allan K. Bertram[1]

[1]Department of Chemistry, University of British Columbia, Vancouver, BC, V6T1Z1, Canada

[2]Institut de Chimie de Clermont-Ferrand, Université Clermont Auvergne, CNRS, Sigma-Clermont, 63000 Clermont-Ferrand, France

*Correspondence to*: Allan Bertram (bertram@chem.ubc.ca)

**Abstract.** A wide range of materials including mineral dust, soil dust, and bioaerosols have been shown to act as ice nuclei in the atmosphere. During atmospheric transport, these materials can become coated with inorganic and organic solutes which may impact their ability to nucleate ice. While a number of studies have investigated the impact of solutes at low concentrations on ice nucleation by mineral dusts, very few studies have examined their impact on non-mineral dust ice nuclei. We studied the effect of dilute $(NH_4)_2SO_4$ solutions (0.05 M) on immersion freezing of a variety of non-mineral dust ice nucleating substances (INSs) including bacteria, fungi, sea ice diatom exudates, sea surface microlayer, and humic substances using the droplet freezing technique. We also studied the effect of $(NH_4)_2SO_4$ solutions (0.05 M) on immersion freezing of several types of mineral dust particles for comparison purposes. $(NH_4)_2SO_4$ had no effect on the median freezing temperature ($\Delta T_{50}$) of nine of the ten tested non-mineral dust materials. There was a small but statistically significant decrease in $\Delta T_{50}$ (-0.43 ± 0.19 °C) for the bacteria *X. campestris* in the presence of $(NH_4)_2SO_4$ compared to pure water. Conversely, $(NH_4)_2SO_4$ increased the median freezing temperature of four different mineral dusts (Potassium-rich feldspar, Arizona Test Dust, Kaolinite, Montmorillonite) by 3 °C to 9 °C and increased the ice nucleation active site density per gram of material ($n_m(T)$) by a factor of ~10 to ~30. This significant difference in the response of mineral dust and non-mineral dust ice nucleating substances when exposed to $(NH_4)_2SO_4$ suggests that they nucleate ice and/or interact with $(NH_4)_2SO_4$ via different mechanisms. This difference suggests that the relative importance of mineral dust to non-mineral dust particles for ice nucleation in mixed-phase clouds could potentially increase as these particles become coated with ammonium sulfate in the atmosphere. This difference also suggests that the addition of $(NH_4)_2SO_4$ (0.05M) to atmospheric samples of unknown composition could potentially be used as an indicator or assay for the presence of mineral dust ice nuclei, although additional studies are still needed as a function of INS concentration to confirm the same trends are observed for different INS concentrations than used here. A comparison with results in the literature does suggest that our results may be applicable to a range of mineral dust and non-mineral dust INS concentrations.

## 1 Introduction

Ice can form homogenously in the atmosphere at temperatures ≲ -35 °C (Koop and Murray, 2016) or heterogeneously at warmer temperatures when an ice nucleating substance (INS) is present to initiate freezing (Murray et al., 2012; Kanji et al., 2017; Hoose and Möhler, 2012). Heterogeneous ice nucleation can take place via several different modes: immersion freezing, deposition nucleation, pore-condensation freezing, and contact freezing (Vali et al., 2015; David et al., 2019). Here we study immersion freezing, which involves the initiation of ice formation by an INS immersed in an aqueous droplet (Vali et al., 2015). This mechanism is thought to dominate ice formation in mixed-phase clouds (Ansmann et al., 2009; Westbrook and Illingworth, 2011).

Atmospheric INSs include mineral dust, soil dust, and bioaerosols (Murray et al., 2012; Kanji et al., 2017; Hoose and Möhler, 2012; Tang et al., 2016). While in the atmosphere, INSs can be transported over long distances and coated with organic and inorganic solutes (Burrows et al., 2009; Fröhlich-Nowoisky et al., 2016; Hinz et al., 2005; Tinsley et al., 2000; McNaughton et al., 2009; Usher et al., 2003; Falkovich et al., 2004). Therefore, to effectively predict ice nucleation in the atmosphere, the effects of solutes on the freezing properties of INSs in the immersion mode need to be determined. A better understanding of the effects of solutes on freezing properties may also lead to a better understanding of the mechanism of heterogeneous ice nucleation in general, which remains highly uncertain (Coluzza et al., 2017). Additionally, if different INS-solute combinations produce known and unique changes in freezing properties, it may be possible to use freezing responses to solute additions as "fingerprints" for different INSs in atmospheric samples, as suggested by Reischel and Vali (1975).

Solutes can decrease the ice nucleating ability of INSs in the immersion mode by lowering the water activity in the solution (i.e. freezing point depression) (Rigg et al., 2013; Koop et al., 2000; Koop and Zobrist, 2009; Zobrist et al., 2008). Solutes can also modify the ice nucleating ability of an INS by interacting with and/or modifying its surface, even at low solute concentrations (< 0.1 M). Several studies have investigated the effects of solutes at low concentrations on the freezing properties of mineral dusts in the immersion mode. Aqueous $NH_3$ and $NH_4^+$ salts at low concentrations improve the ice nucleation ability of feldspars, micas, gibbsite, quartz, and kaolinite, and have little to no effect on the ice nucleation ability of amorphous silica particles (Kumar et al., 2019a, b, 2018; Reischel and Vali, 1975; Whale et al., 2018). In some cases, potassium salts improve the ice nucleation ability of feldspars depending on the concentration of the salts and the freezing temperature (Yun et al., 2020; Perkins et al., 2020). Lithium iodide was found to increase the freezing temperature of kaolinite particles in one study (Reischel and Vali, 1975), but not in a more recent study (Ren et al., 2020). Other inorganic salts including, NaOH and NaCl, decrease the freezing temperatures of some types of mineral dust (Kumar et al., 2019a, 2018, 2019b; Reischel and Vali, 1975; Whale et al., 2015). Inorganic acids either decrease the ice nucleation ability of mineral dust particles or have little effect, depending on the type of acid, exposure time, concentration of the acid, and type

of mineral dust (Kumar et al., 2018; Burkert-Kohn et al., 2017; Sullivan et al., 2010b; Tobo et al., 2012; Augustin-Bauditz et al., 2014; Wex et al., 2014; Sullivan et al., 2010a; Link et al., 2020). On the other hand, organic solutes have often been found to have no effect on the ice nucleating ability of mineral dust particles (Zobrist et al., 2008; Koop and Zobrist, 2009; Tobo et al., 2012; Wex et al., 2014; Kanji et al., 2019).

In comparison to mineral dust, there have only been a small number of studies that have investigated the effect of solutes at low concentrations on non-mineral dust INSs. Reischel and Vali (1975) studied the effects of a range of inorganic salts on the freezing properties of leaf derived nuclei and found only small changes (less than 1.5 °C) in the freezing temperatures of this INS in the presence of each of the tested solutes. Whale et al. (2018) studied the effects of $(NH_4)_2SO_4$ and NaCl on the ice nucleation ability of humic acid and found no significant change in freezing temperature in the presence of either solute.

Attard et al. (2012) studied the effect of pH on the freezing properties of several *Pseudomonas* strains, and found that acidic solutions decreased the ice nucleation activity of the *Pseudomonas* strains studied. Koop and Zobrist (2009) studied the effects of the solutes $(NH_4)_2SO_4$, glucose, $H_2SO_4$, and PEG400 on the freezing properties of Snomax (a commercial product for artificial snow production made from components of *Pseudomonas syringae*) and found no effect of the solutes on the freezing temperature other than freezing point depression. Chernoff and Bertram (2010) studied the effects of $H_2SO_4$

coatings on the freezing properties of Snomax and similarly found that the coating caused no significant change in the ice nucleating properties other than freezing point depression. Amato et al. (2015) injected *Pseudomonas syringae* suspensions in $(NH_4)_2SO_4$ into a cloud simulation chamber and observed a slight decrease in the ice nucleating activity compared to the cells in water, although these data were not corrected for freezing point depression by the solute. Weng et al. (2016) studied the effects of the cryoprotectants ethylene glycol, propylene glycol, and trehalose on *Pseudomonas syringae* and found no

effect of the solutes on freezing temperature other than freezing point depression. Desnos et al. (2020) studied the effects of the cyroprotectant $Me_2SO$ on the freezing properties of Snomax and observed a decrease in the ice nucleation activity that was greater than that produced by freezing point depression. Schwidetzky et al. (2021) studied the effect of the inorganic salts NaCl, $NH_4Cl$, NaSCN, $MgSO_4$ on Snomax and found that NaSCN and $NH_4Cl$ decreased the freezing temperature of Snomax, NaCl had no effect on the freezing temperature, and $MgSO_4$ increased the freezing temperature of Snomax.

    To expand on the limited studies mentioned above, we investigated the effect of $(NH_4)_2SO_4$ at a low concentration (0.05 M) on the freezing properties of several types of non-mineral dust INSs of atmospheric relevance. $(NH_4)_2SO_4$ was chosen because it is a common inorganic solute in the atmosphere. A concentration of 0.05 M was chosen because it is relevant for mixed phase clouds in the atmosphere. Because $(NH_4)_2SO_4$ causes an increase in the ice nucleation ability of most mineral

dust particles even at low concentrations, we investigated whether it would have a similar effect on non-mineral dust INSs in the immersion mode. If $(NH_4)_2SO_4$ has little to no effect on the freezing properties of non-mineral dust INSs, then a change in freezing temperatures of atmospheric samples in response to the addition of low concentrations of $(NH_4)_2SO_4$ could potentially be used to identify the presence of mineral dust INSs in atmospheric samples.

## 2 Experimental

 **2.1 INS suspensions**

We investigated the effect of $(NH_4)_2SO_4$ on the ice nucleating ability of a range of non-mineral dust INSs of atmospheric relevance. INSs studied include bacteria, fungi, exudates from sea ice diatoms collected in Antarctica, INSs from the sea surface microlayer collected in the Arctic, and humic substances. For comparison purposes, we also investigated the effect of $(NH_4)_2SO_4$ on the ice nucleating ability of four types of mineral dust (Arizona Test Dust, K-rich feldspar, montmorillonite, and kaolinite). The effect of $(NH_4)_2SO_4$ on the ice nucleating ability of Arizona Test Dust, K-rich feldspar, and kaolinite, has been studied before (Kumar et al., 2018; Whale et al., 2018; Kumar et al., 2019b), but the effect on montmorillonite has not. Details of the studied INSs and how suspensions of the INSs were prepared are given below.

### 2.1.1 Bacteria

*Pseudomonas syringae* (*P. syringae*) has been identified in the atmosphere and is an extremely effective ice nuclei with ice nucleation temperatures as high as -3 to -4 °C (Ahern et al., 2007; Amato et al., 2007; Lindemann et al., 1982; Maki et al., 1974). Strain 31R1 was provided by S. Lindow (Department of Plant and Microbial Biology, University of California, Berkley, U.S.A.). The ice nucleation ability of Strain 31R1 has been studied previously (Lindow et al., 1989; Möhler et al., 2008). This strain was grown in nutrient broth for 5 days at 26 °C shaking at 225 rpm. After growth, the suspension was diluted in ultrapure water to an optical density of ~0.06 at a wavelength of 600 nm, which should be approximately equal to a cell number concentration of ~$1 \times 10^8$ cells mL$^{-1}$. Ultrapure water refers to distilled water purified by a Millipore system to a resistivity 18.2 MΩ·cm at 25 °C. A blank for the *P. syringae* sample was prepared using the nutrient broth growth media at the same dilution factor as in the INS suspension.

*Xanthomonas campestris* (*X. campestris*) has been identified in cloud droplets (Vaïtilingom et al., 2012) and its ice nucleating ability in the immersion mode has been previously characterized (Joly et al., 2013; Kim et al., 1987). Strain 32b-52 was provided by P. Amato (Institut de Chimie de Clermont-Ferrand CNRS-Clermont University, Aubière, France). The ice nucleation ability of this strain has been studied previously (Joly et al., 2013). This strain was grown in R2A broth for 2 days at 17 °C shaking at 225 rpm. After growth, the suspension was diluted to an optical density of ~0.06 which should be equivalent to a cell number concentration of ~$1 \times 10^8$ cells mL$^{-1}$. A blank for the *X. campestris* sample was prepared using the R2A broth growth media at the same dilution factor as in the INS suspension.

Snomax is a commercial product for artificial snow production made of proteins from *P. syringae* cells. The freezing ability of Snomax in the immersion mode has been characterized in several previous studies (Koop and Zobrist, 2009; Möhler et al., 2008; Whale et al., 2015; Wex et al., 2015). A 0.1 wt% suspension of Snomax (Johnson Controls Snow) was prepared in ultrapure water leading to a cell concentration of ~$7 \times 10^8$ cells mL$^{-1}$ (Möhler et al., 2008; Koop and Zobrist, 2009).

### 2.1.2 Fungi

*Fusarium acuminatum* (*F. acuminatum*) and *Fusarium avenaceum* (*F. avenaceum*) have been identified in the atmosphere and have been found to be effective INSs in the immersion mode (Amato et al., 2007; Hasegawa et al., 1994; Pouleur et al., 1992; Richard et al., 1996; Seifi et al., 2014). Cultures were obtained from the American Type Culture Collection (ATCC catalogue numbers 60315 and 200466 for *F. acuminatum* and *F. avenaceum,* respectively) and grown in potato dextrose broth for 3-4 days at approximately 21 °C shaking at 50 rpm. After growth, the suspensions were diluted in ultrapure water to an optical density of ~0.1 at a wavelength of 530 nm which should be equivalent to a cell number concentration on the order of $10^6$ cells mL$^{-1}$ (Petrikkou et al., 2001). The specific relationship between cell number concentration and optical density at 530 nm has not been standardized for the fungi studied here, therefore the cell number concentration of $10^6$ cells mL$^{-1}$ should be considered highly uncertain. A blank for the fungal samples was prepared using the potato dextrose broth growth media at the same dilution factor as in the INS suspension.

### 2.1.3 Exudates from sea-ice diatoms collected in Antarctica

Recent studies have shown that marine diatoms and their exudates can act as effective ice nuclei (Alpert et al., 2011; Knopf et al., 2011; Ladino et al., 2016; Wang et al., 2015; Wilson et al., 2015; Ickes et al., 2020). In a recent study we showed that exudates from sea-ice diatoms collected in Antarctica also contain INSs (Xi et al., 2021). After melting of sea ice, these INSs can be transferred to the atmosphere by the bubble bursting mechanism (Blanchard, 1964). Here we used a sample of exudates from dense sea-ice diatom communities consisting of material collected at Tent Island (77°41'7.90"S, 166°25'14.62"E) and Inaccessible Island (77°40′S, 166°22′E) in November 1998 (Raymond, 2000). The sample was originally collected to study the ice binding properties of exudates from sea-ice diatoms (Raymond, 2000). Chunks of brown ice containing sea-ice diatoms were collected from the underside of sea ice. Antarctic sea ice at the ice-water interface can become brown as a result of the growth of dense diatom communities (Grossi et al., 1987), hence brown ice was specifically collected. The collected brown ice was rich in diatoms and contained few other organisms based on visual inspection with a dissecting microscope. The chunks of brown ice were melted and the diatom cells were allowed to settle overnight. The supernatant of the sample, which contained the diatom exudates, was then subjected to two cycles of ice-affinity purification to isolate ice binding substances in the supernatants (Raymond and Fritsen, 2001; Raymond, 2000). In each cycle of ice affinity purification, the sample containing ice binding materials was half frozen at -5 °C and the ice was harvested (Raymond, 2000). This process, which was carried out directly after sampling, concentrated ice binding materials in the sample. After purification, the sample was stored at -20 °C until use. For the Antarctic diatom exudate sample, we used ultrapure water as the blank.

### 2.1.4 Sea surface microlayer collected in the Arctic

The sea surface microlayer (SML), which has a thickness of less than 1 mm, is defined as the interface between the ocean and atmosphere (Liss and Duce, 1997) and has been shown to contain ice nucleating substances (Wilson et al., 2015; Irish et al., 2017, 2019; Mccluskey et al., 2018; Wolf et al., 2020). INSs within this interface can be lofted into the atmosphere via the bubble bursting mechanism (Blanchard, 1964; Ickes et al., 2020; Mccluskey et al., 2018; DeMott et al., 2016). In a previous study, we showed that SML samples collected in the eastern Arctic during the period of July and August 2016 contained INSs (Irish et al., 2019). These INSs were most likely heat-labile biological materials of < 0.2 μm in size (Irish et al., 2019). Two of the samples investigated in the study (microlayer sampling station 2 collected at 67°23.466' N, 063°22.067' W and microlayer sampling station 7 collected at 77°47.213' N, 076°29.841' W) were also used here. The microlayer samples were collected with the glass plate technique, where a glass plate was immersed into the water and withdrawn slowly and the microlayer that adhered to the glass plate was scraped off into a collection vessel as detailed in Irish et al. (2019). After collection samples were stored at -80 °C. For the sea surface microlayer samples, we used ultrapure water as the blank.

### 2.1.5 Humic and fulvic acid

Humic acid and fulvic acid are mixtures of macromolecular organic compounds isolated from soils and water. Humic like substances contain multiple organic compounds that are abundant in soils and have been identified in atmospheric aerosol (Graber and Rudich, 2006; White, 2009). The ice nucleating abilities of humic and fulvic acid have been characterized in previous studies (Knopf et al., 2010; O'Sullivan et al., 2014; Pratt et al., 2009; Shilling et al., 2006; Borduas-Dedekind et al., 2019). A Suwanee River fulvic acid standard and a Leonardite humic acid standard were obtained from the International Humic Substances Society, and 0.1 wt % solutions were prepared using ultrapure water. For humic and fulvic acid, we used ultrapure water as the blank.

### 2.1.6 Mineral dust

Potassium-rich feldspar (K-rich feldspar) is an important component of atmospheric mineral dust. The ice nucleation ability of K-rich feldspar has been quantified previously (Peckhaus et al., 2016a; Atkinson et al., 2013; Whale et al., 2018; Yun et al., 2020; Harrison et al., 2016). A K-rich feldspar sample was obtained from the Pacific Museum of Earth, University of British Columbia, and ground into a powder using a mortar and pestle. We have characterized the ice nucleating ability of this specific power in a previous study (Yun et al., 2020). The mineralogy of the powder was 85% microcline ($KAlSi_3O_8$) and 15% albite ($NaAlSi_3O_8$) as determined by X-ray diffraction measurements. The specific surface area of the particles after grinding was 0.75 $m^2$ $g^{-1}$ based on Brunauer-Emmet-Teller (BET) nitrogen adsorption method (Yun et al., 2020). A 0.1 wt% suspension of this K-feldspar powder was prepared in ultrapure water and stirred overnight before freezing experiments to ensure even particle distribution. For the mineral dust samples, concentrations were adjusted to ensure that freezing occurred

at temperatures warmer than the blanks (>20 °C). As a result, some mineral dust suspensions (kaolinite and montmorillonite suspensions) were prepared at different concentrations than used for K-rich feldspar (see below). For all the mineral dust samples, we used ultrapure water as a blank.

Arizona Test Dust (ATD) is a commercially available mineral dust that is often used as a proxy for mineral dust in the atmosphere. The ice nucleating ability of ATD has been characterized in several studies (Zobrist et al., 2008; Whale et al., 2018; Wheeler et al., 2015; Perkins et al., 2020; Kanji and Abbatt, 2010; Knopf and Koop, 2006). A 0.1 wt % suspension of Arizona Test Dust (Powder Technology Inc, 0-3 µm size fraction) was prepared in ultrapure water and stirred overnight before freezing experiments to ensure even particle distribution.


Kaolinite is another mineral abundant in the atmosphere (Tang et al., 2016; Broadley et al., 2012; Kandler et al., 2007; Delany et al., 1967; Glaccum and Prospero, 1980). The ice nucleation ability of kaolinite has been previously characterized (Ren et al., 2020; Hoose and Möhler, 2012; Kumar et al., 2019b; Chernoff and Bertram, 2010; Pinti et al., 2012; Wex et al., 2014; Lüönd et al., 2010). A Kaolinite sample (KGa-1b) was obtained from the Clay Minerals Society, Purdue University.
The mineralogy of the sample was 94.7% kaolinite ($Al_2Si_2O_5(OH)_5$), 3.6% anatase ($TiO_2$), and 1.7% feldspar (orthoclase) as determined by X-ray diffraction measurements. A 1 wt% suspension of kaolinite was prepared in ultrapure water and stirred overnight before freezing experiments to ensure even particle distribution.

Montmorillonite is a mineral found in high abundance in African and Asian clay dusts (Tang et al., 2016; Glaccum and
Prospero, 1980; Prospero, 1999; Ganor, 1991). The ice nucleation ability of montmorillonite has been previously characterized (Eastwood et al., 2008; Salam et al., 2007; Hoose and Möhler, 2012; Chernoff and Bertram, 2010; Kulkarni et al., 2014; Kaufmann et al., 2016; Atkinson et al., 2013). A montmorillonite sample was obtained from ThermoFisher Scientific, -200 Mesh Powder. The sample was mined and crushed to 200 Mesh Powder and was not chemically processed. A 1 wt% suspension of montmorillonite was prepared in ultrapure water and stirred overnight before freezing experiments to
ensure even particle distribution.

## 2.2 Droplet freezing experiments

The effects of $(NH_4)_2SO_4$ on the ice nucleating abilities of the INS samples were determined using the droplet freezing technique (Fig. 1a) (Whale et al., 2015; Vali, 1971). We have used this technique previously to study the ice nucleating properties of mineral dust, sea surface microlayer and bulk sea water, and exudates from diatoms (Xi et al., 2021; Yun et al.,
2020; Irish et al., 2019). Three hydrophobic glass slides (Hampton Research HR3-239) were rinsed with ultrapure water, dried with nitrogen gas, and placed on a cold stage (Grant Asymptote EF600 Cryocooler). Twenty 1 µL drops from an INS suspension were placed onto each slide using a micropipette. A second 1 µL drop of either 0.1 M $(NH_4)_2SO_4$ or ultrapure water was added to each sample droplet with a micropipette, bringing the total volume of each droplet to 2 µL. The 0.1 M

(NH₄)₂SO₄ solution (Fischer Scientific $(NH_4)_2SO_4$, ACS grade) was prepared in ultrapure water. A chamber with an attached digital camera was placed over the droplets to isolate them from the ambient air. A small flow of ultrapure nitrogen gas (0.2 L min⁻¹) was passed through the chamber to prevent condensation of water on the slides during cooling. The flow did not cause evaporation of the droplets or affect the freezing temperature of the droplets (Whale et al., 2015). The temperature of the cold stage was decreased at a rate of 3 °C min⁻¹ until all sample droplets were frozen. The digital camera attached to the chamber recorded videos of the freezing process. Examples of images recorded are shown in Fig. 1b-c. The video and temperature data from the cold stage were processed using a MATLAB script to determine the freezing temperature of each droplet (Xi et al., 2021). The uncertainty in the cold stage temperature measurements was approximately ±0.25 °C according to the manufacturer specifications, which was verified by measuring the melting point of water and dodecane and comparing the measured melting points with literature values (Lide, 2001).

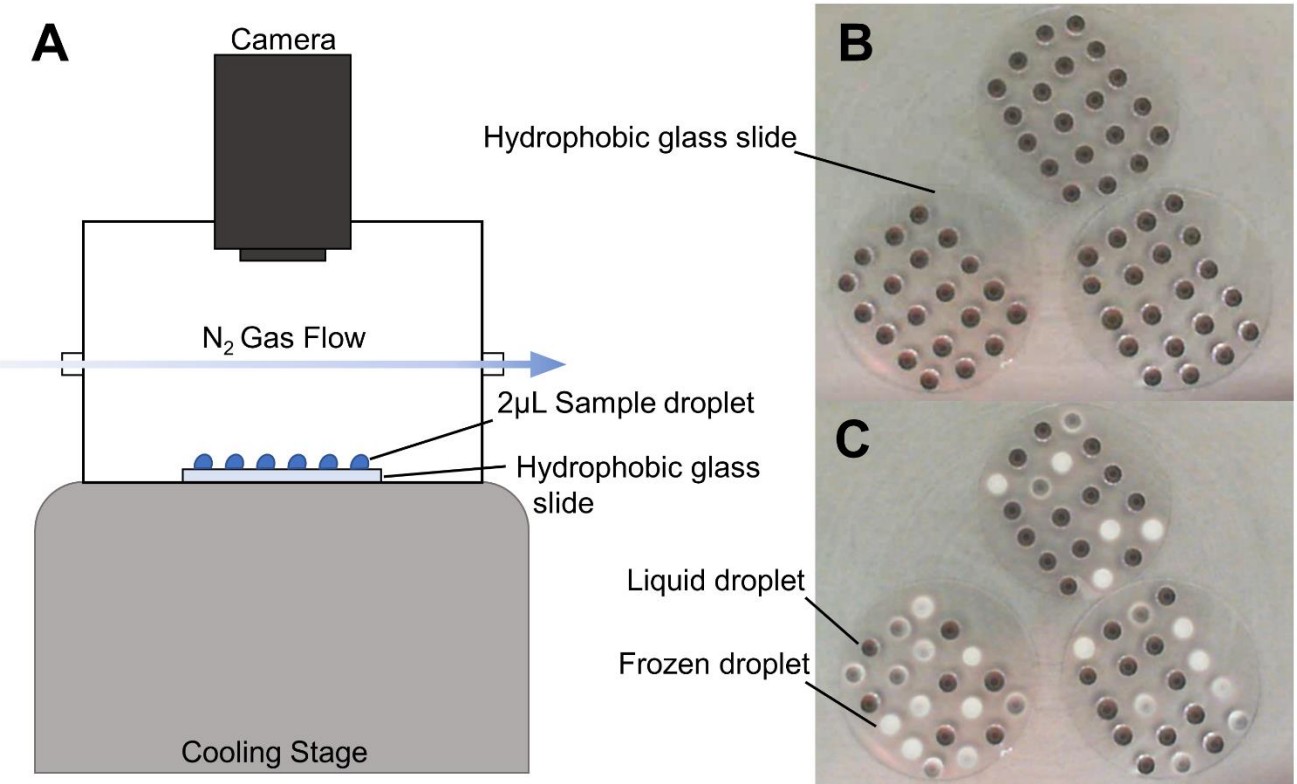

**Figure 1. Experimental setup for droplet freezing experiments. A) Schematic of droplet freezing setup. Images taken by the digital camera during freezing experiments with *F. avenaceum*, taken at approximately B) -3 °C and C) -11.5 °C, respectively. The diameter of each glass slide is 18mm.**

In cases where $(NH_4)_2SO_4$ was added to the droplets in the freezing experiments, the freezing temperatures of the droplets were corrected for the freezing point depression caused by the solute using the following equation (Atkins and de Paula, 2014):

$$\Delta T_f = iK_f m_{solute} \tag{1}$$

where $\Delta T_f$ is the freezing point depression, $i$ is the van't Hoff factor, $K_f$ is the cryoscopic constant of water (1.86 °C·kg mol$^{-1}$),

and $m_{solute}$ is the molality of $(NH_4)_2SO_4$ in the 2 μL droplets. For an $(NH_4)_2SO_4$ concentration of 0.05 M (the concentration of $(NH_4)_2SO_4$ in the 2 μL droplets) the calculated freezing point depression was 0.28 °C. Freezing temperatures reported here (both samples and blanks) have been corrected for the freezing point depression caused by the addition of $(NH_4)_2SO_4$.

From the freezing temperatures of individual droplets, we determined the fraction of droplets frozen as a function of temperature. For the majority of samples, we also have calculated ice nucleation active site densities per gram of material as a function of temperature, $n_m(T)$ (g$^{-1}$), using the following equations:

$$[INS(T)] = \frac{-\ln\left(N_u(T)/N_o\right)}{V} \tag{2}$$

and

$$n_m(T) = [INS(T)]\frac{V}{m} \tag{3}$$

Where $[INS(T)]$ (L$^{-1}$) is the concentration of INS per liter of solution as a function of temperature, $N_u(T)$ is the number of droplets in an experiment that are unfrozen at temperature $T$, $N_o$ is the total number of droplets in an experiment, $V$ is the volume of an individual droplet, and $m$ is the average mass of material per droplet. For the cultured bacterial and fungal samples, the mass of material in the cultures were calculated using the approximate cell concentrations from optical density

measurements and assuming a cell density of 1.1 g cm$^{-3}$ (Bakken and Olsen, 1983) and cell volume of 1.595 μm$^3$ (Buchanan and Gibbons, 1974), 0.8082 μm$^3$ (Sharma et al., 2014), 440.7 μm$^3$ (Burgess et al., 1993), and 842.3 μm$^3$ (Yli-Mattila et al., 2018) for *P. syringae*, *X. campestris*, *F. acuminatum*, and *F. avenaceum,* respectively. Note that these calculations involve multiple approximations, particularly for the fungal samples where the cell concentrations and volumes are highly uncertain, therefore these $n_m(T)$ values should be taken as estimates.


An average mass of material per droplet could not be determined for the sea surface microlayer and Antarctic diatom exudate samples because they are environmental samples of unknown composition, hence $n_m(T)$ values could not be calculated. For these samples we report $[INS(T)]$ values.

Fraction frozen, $n_m(T)$, and $[INS(T)]$ values are presented as averages of the three to six replicates for a given INS suspension at a given temperature. Error bars represent the 95% confidence intervals of the fraction frozen, $n_m(T)$, or

$[INS(T)]$ values at that temperature. The 95% confidence interval should account for both sample-to-sample variability and nucleation statistics. Average values of $[INS(T)]$ were also calculated for the blanks using Eq. (3) and subtracted from the samples prior to reporting $n_m(T)$ and $[INS(T)]$ values for the samples.

## 3. Results and Discussion

### 3.1 Effects of (NH₄)₂SO₄ on ice nucleation ability of non-mineral dust INSs

Shown in Fig. 2-7 are the fraction frozen curves for the 2 μL droplets containing non-mineral dust INSs with and without 0.05 M (NH₄)₂SO₄. Also shown in Fig. 2-7 are fraction frozen curves for laboratory blanks. For the bacteria and fungi grown in the laboratory, blanks correspond to the fraction frozen curves for the growth medium at the same dilution factor as in the INS suspensions. For all other cases, blanks correspond to fraction frozen curves of 2 μL droplets of ultrapure water. On average, the fraction frozen curves for the INSs were shifted to warmer temperatures compared to the fraction frozen curves for the blanks (Fig. 2-7). As a result, we concluded the INSs were responsible for ice nucleation in our experiments.

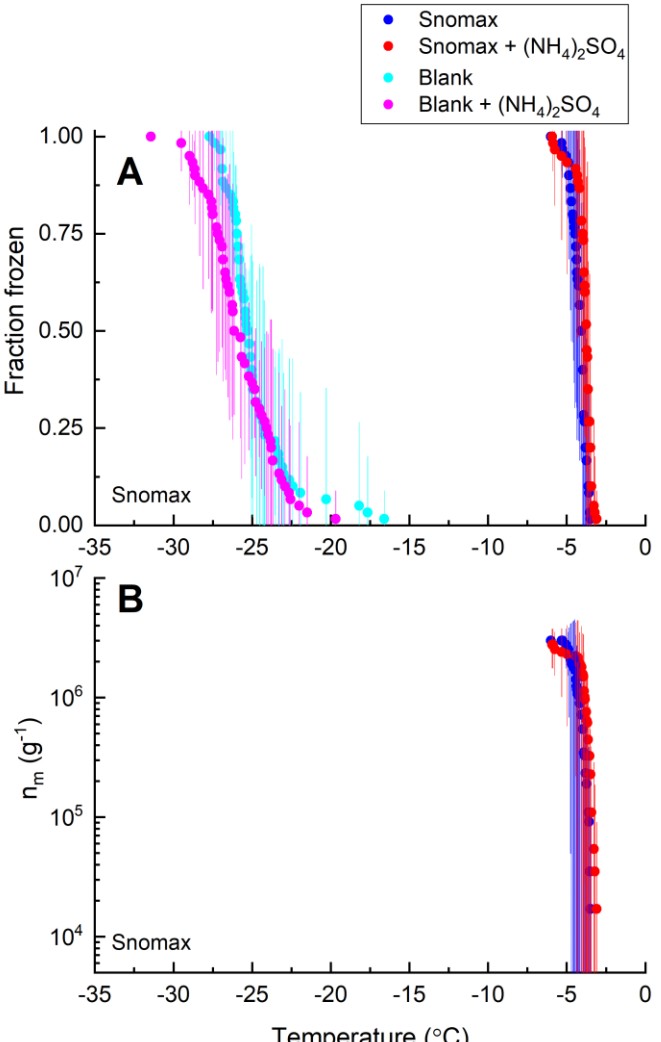

**Figure 2. Fraction of droplets frozen (A) and ice nucleation active site densities $n_m(T)$ (B) for samples of Snomax with (red) and without (blue) 0.05 M (NH₄)₂SO₄ compared with an ultrapure water blank with (pink) and without (cyan) 0.05 M (NH₄)₂SO₄. Freezing data (both samples and blanks) have been corrected for freezing point depression by 0.05 M (NH₄)₂SO₄. Error bars represent the 95% confidence interval of fraction frozen or $n_m(T)$ at temperature T calculated using the Student's T distribution.**


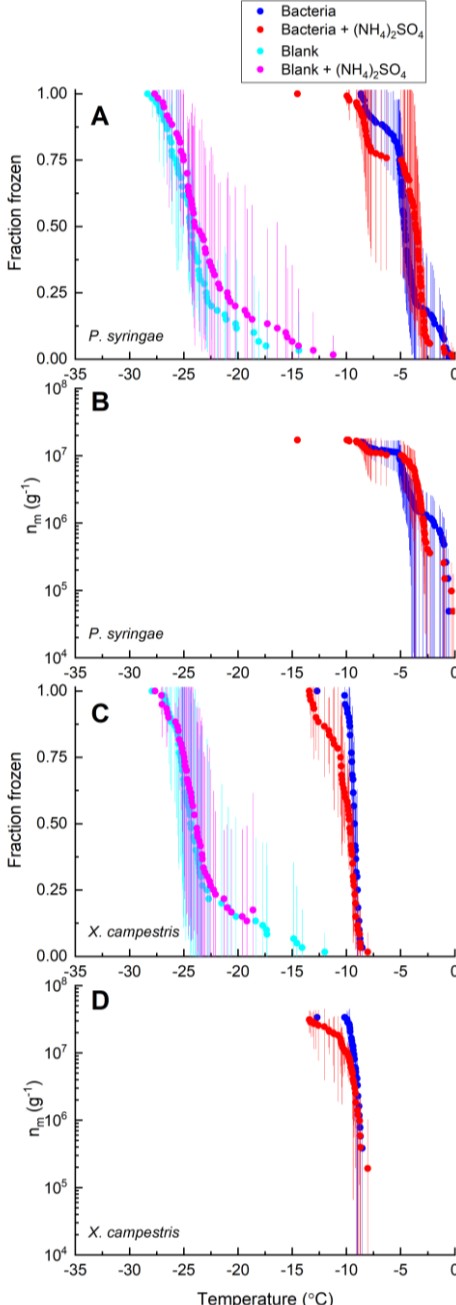

**Figure 3.** Fraction of droplets frozen (A, C) and ice nucleation active site densities $n_m(T)$ (B, D) for samples of *P. syringae* (A, B) and *X. campestris* (C, D) with (red) and without (blue) 0.05 M $(NH_4)_2SO_4$ compared with the respective growth media blank with (pink) and without (cyan) 0.05 M $(NH_4)_2SO_4$. Freezing data (both samples and blanks) have been corrected for freezing point depression by 0.05 M $(NH_4)_2SO_4$. Error bars represent the 95% confidence interval of fraction frozen or $n_m(T)$ at temperature T calculated using the Student's T distribution. For *P. syringae*, two drops out of 120 froze close to the blanks, which is inconsistent with previous studies reported in the literature. We assume the low freezing temperature of these two droplets is because these two drops did not contain any cells. Hence, we removed these two data points from our analysis.

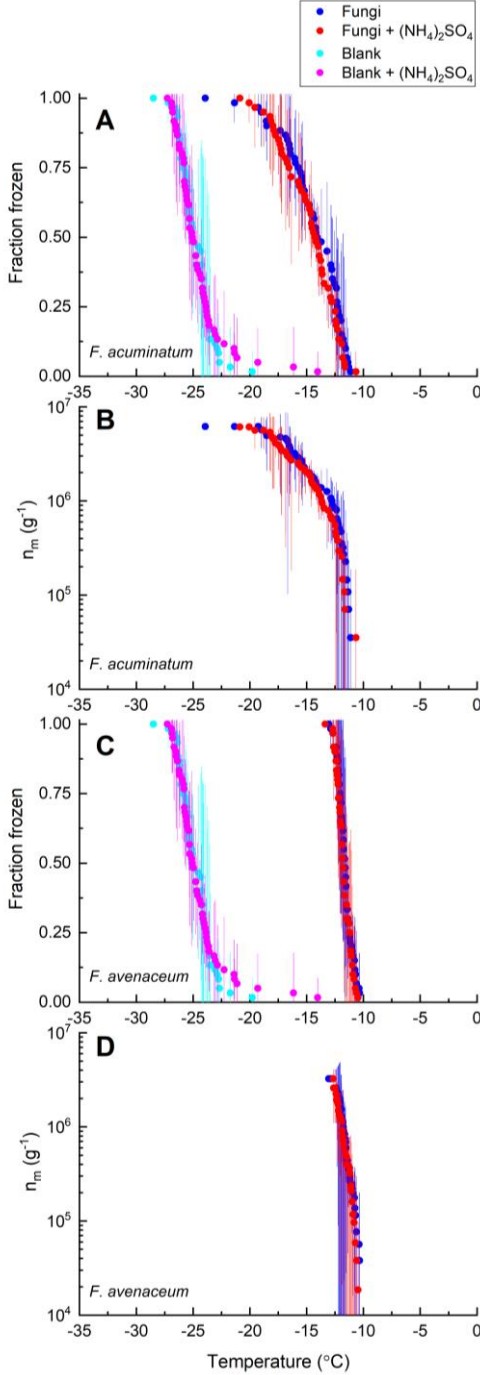

**Figure 4. Fraction of droplets frozen (A, C) and ice nucleation active site densities $n_m(T)$ (B, D) for samples of *F. acuminatum* (A,**
**B) and *F. avenaceum* (C, D) with (red) and without (blue) 0.05 M (NH$_4$)$_2$SO$_4$ compared with the respective growth medium with**
**(pink) and without (cyan) 0.05 M (NH$_4$)$_2$SO$_4$. Freezing data (both samples and blanks) have been corrected for freezing point**
**depression by 0.05 M (NH$_4$)$_2$SO$_4$. Error bars represent the 95% confidence interval of fraction frozen or $n_m(T)$ at temperature T**
**calculated using the Student's T distribution.**

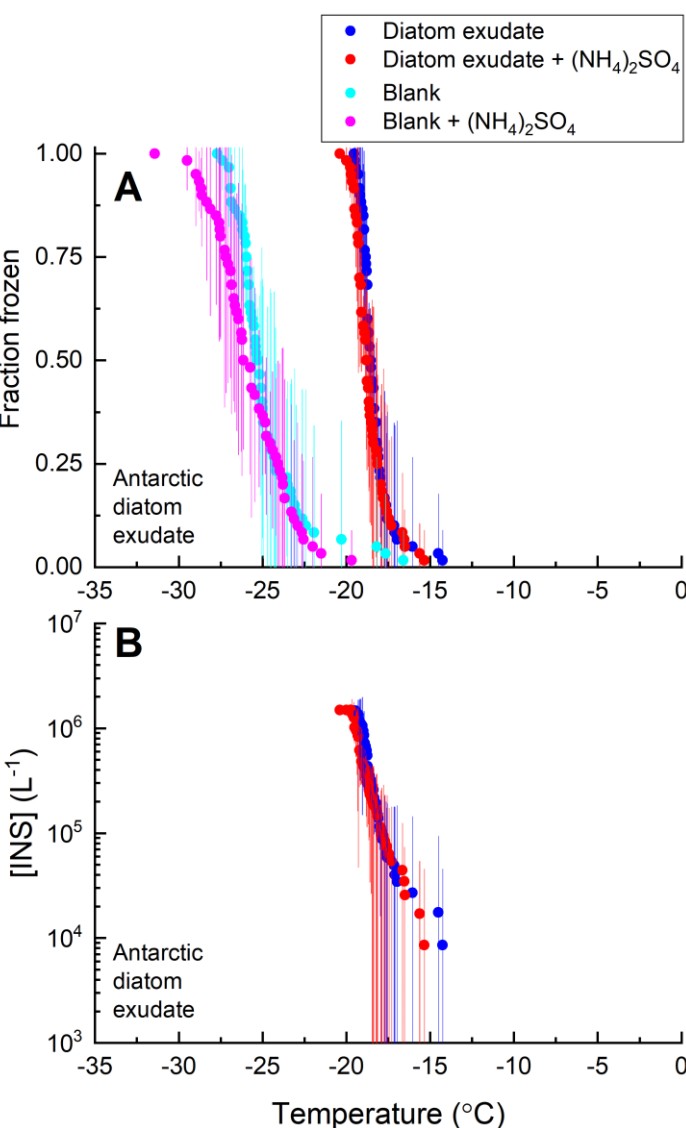

**Figure 5.** Fraction of droplets frozen (A) and the concentration of INS per liter of solution [$INS(T)$] (B) for samples of Antarctic diatom exudates with (red) and without (blue) 0.05 M (NH4)2SO4 compared with an ultrapure water blank with (pink) and without (cyan) 0.05 M (NH4)2SO4. Freezing data (both samples and blanks) have been corrected for freezing point depression by 0.05 M (NH4)2SO4. Error bars represent the 95% confidence interval of fraction frozen or [$INS(T)$] at temperature T calculated using the Student's T distribution.

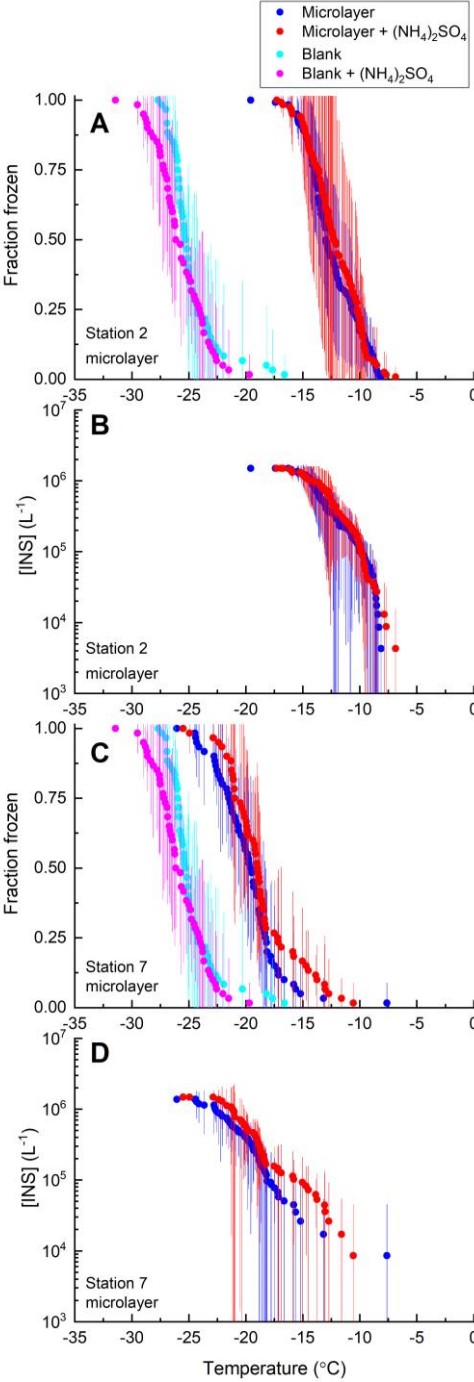

**Figure 6. Fraction of droplets frozen (A, C) and the concentration of INS per liter of solution $[INS(T)]$ (B, D) for samples of Arctic sea surface microlayer from Station 2 (A, B) and Station 7 (C, D) with (red) and without (blue) 0.05 M (NH₄)₂SO₄ compared with an ultrapure water blank with (pink) and without (cyan) 0.05 M (NH₄)₂SO₄. Freezing data (both samples and blanks) have been corrected for freezing point depression by 0.05 M (NH₄)₂SO₄. Error bars represent the 95% confidence interval of fraction frozen or $[INS(T)]$ at temperature T calculated using the Student's T distribution.**

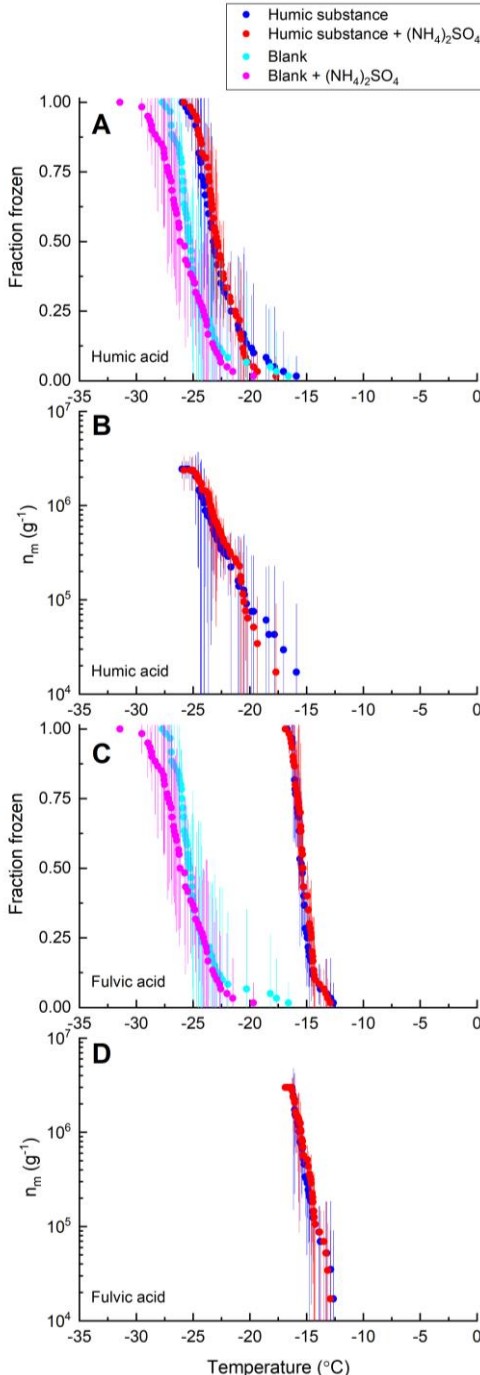

**Figure 7. Fraction of droplets frozen (A, C) and ice nucleation active site densities $n_m(T)$ (B, D) for samples of humic acid (A, B) and fulvic acid (C, D) with (red) and without (blue) 0.05 M $(NH_4)_2SO_4$ compared with an ultrapure water blank with (pink) and without (cyan) 0.05 M $(NH_4)_2SO_4$. Freezing data (both samples and blanks) have been corrected for freezing point depression by 0.05 M $(NH_4)_2SO_4$. Error bars represent the 95% confidence interval of fraction frozen or $n_m(T)$ at temperature T calculated using the Student's T distribution**

For all the non-mineral dust INSs, the fraction frozen curves in the presence of $(NH_4)_2SO_4$ overlap with the fraction frozen curves without $(NH_4)_2SO_4$ (Fig. 2-7). Furthermore, the shapes of the fraction frozen curves for droplets containing non-mineral dust INSs were very similar with and without $(NH_4)_2SO_4$. To better compare the effect of $(NH_4)_2SO_4$ on different INSs, we calculated the change in the median freezing temperature ($\Delta T_{50}$) due to the presence of $(NH_4)_2SO_4$ using the following equation:

$$\Delta T_{50} = T_{50,(NH_4)_2SO_4} - T_{50,No\ (NH_4)_2SO_4} \tag{4}$$

where $T_{50,(NH_4)_2SO_4}$ is the median freezing temperature of the droplets containing an INS and $(NH_4)_2SO_4$ (after correcting for freezing point depression) and $T_{50,No\ (NH_4)_2SO_4}$ is the median freezing temperature of droplets containing an INS without $(NH_4)_2SO_4$. $T_{50,(NH_4)_2SO_4}$ and $T_{50,No\ (NH_4)_2SO_4}$ were averages of all replicates for a given INS suspension. Uncertainties in $\Delta T_{50}$ were calculated from the 95% confidence intervals for $T_{50,(NH_4)_2SO_4}$ and $T_{50,No\ (NH_4)_2SO_4}$. $\Delta T_{25}$ and $\Delta T_{75}$ values were calculated similarly and correspond to the change in the freezing temperatures evaluated at a fraction frozen of 0.25 and 0.75, respectively.

For eight of the ten non-mineral dust INSs (Snomax, *P. syringae*, *F. acuminatum*, *F. avenaceum*, SML from station 2, SML from station 7, humic acid, and fulvic acid) $\Delta T_{50}$, $\Delta T_{25}$, and $\Delta T_{75}$ values were all less than the uncertainty in the measurements (95% confidence interval), consistent with no change in freezing temperature upon adding $(NH_4)_2SO_4$ (Fig. 8). For *X. campestris*, a small statistically significant negative $\Delta T_{50}$ value was observed ($\Delta T_{50}$ = -0.43 ± 0.19). For *X. campestris* and the Antarctic diatom exudates, small statistically significant negative $\Delta T_{75}$ values were observed ($\Delta T_{75}$ = -1.27 ± 1.13 and -0.40 ± 0.34 respectively). In no case was a statistically significant positive $\Delta T_{50}$, $\Delta T_{25}$, or $\Delta T_{75}$ observed, consistent with $(NH_4)_2SO_4$ either having no effect on or decreasing the ice nucleation ability of non-mineral dust INSs.

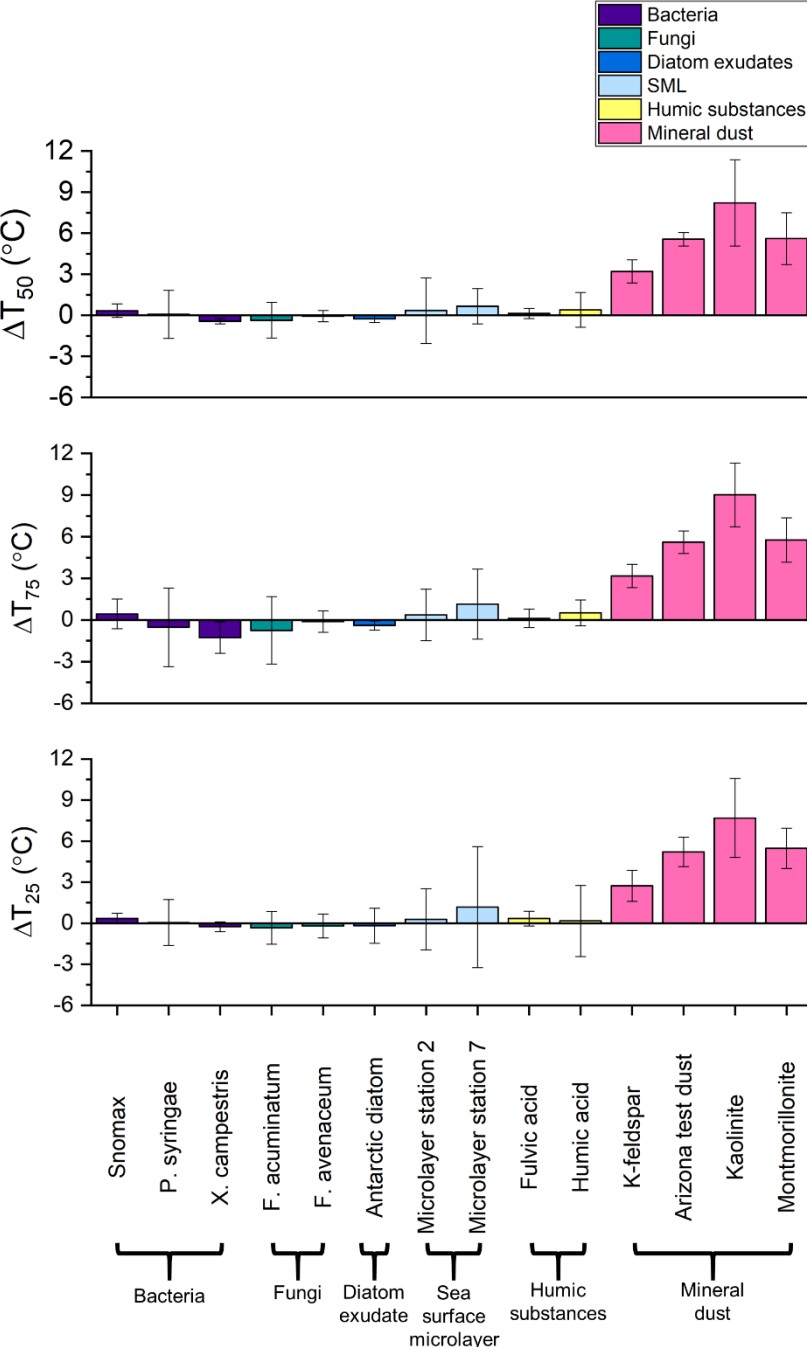

**Figure 8. Change in the temperature at which 50% ($\Delta T_{50}$), 75% ($\Delta T_{75}$), and 25% ($\Delta T_{25}$) of sample droplets were frozen between samples with $(NH_4)_2SO_4$ added and samples without $(NH_4)_2SO_4$ added. Error bars represent the 95% confidence interval calculated using the Student's T distribution. Freezing data have been corrected for freezing point depression by 0.05 M $(NH_4)_2SO_4$.**

Also shown in Fig. 2-7 are the $n_m(T)$ values or $[INS(T)]$ values for the non-mineral dust INSs with and without 0.05 M (NH$_4$)$_2$SO$_4$. Similar to the fraction frozen curves, the $n_m(T)$ and $INS(T)$ values in the presence of (NH$_4$)$_2$SO$_4$ were very similar to those without (NH$_4$)$_2$SO$_4$. Also like the fraction frozen curves, the shapes of the $n_m(T)$ and $[INS(T)]$ curves were very similar with and without (NH$_4$)$_2$SO$_4$.

A few previous studies have also investigated the effect of (NH$_4$)$_2$SO$_4$ on non-mineral INSs (specifically Snomax, humic acid and leaf derived nuclei). The results from these previous studies are summarized in Table S1. We have only included results up to 0.1 M (NH$_4$)$_2$SO$_4$, since this is the range most relevant for our experiments. For comparison purposes we have also summarized our results for Snomax and humic acid in Table S1. The previous experiments include measurements with different concentrations of (NH$_4$)$_2$SO$_4$, different concentrations of non-mineral INSs, and different freezing temperature regimes. In no case was ice nucleation enhanced by the addition of (NH$_4$)$_2$SO$_4$, consistent with our results.

The data shown in Table S1 suggest that (NH$_4$)$_2$SO$_4$ has no impact on non-mineral INSs for a range of (NH$_4$)$_2$SO$_4$ concentrations and a range of non-mineral INS concentrations, at least for some types of non-mineral INSs. For example, Koop and Zobrist, (2009) performed DSC experiments over a range of (NH$_4$)$_2$SO$_4$ concentrations and at a very different concentration of Snomax than in our experiments but observed the same result as us (i.e. no impact on ice nucleation) (Table S1). Similarly, Whale et al., (2018) performed experiments at a higher concentration of humic acid and a lower concentration of (NH$_4$)$_2$SO$_4$ than our experiments but observed the same result of no impact on ice nucleation (Table S1). Reischel and Vali, (1975) studied ice nucleation by leaf derived nuclei in the presence of three different concentrations of (NH$_4$)$_2$SO$_4$ and observed little to no impact on ice nucleation at each concentration (Table S1).

## 3.2 Effects of (NH$_4$)$_2$SO$_4$ on ice nucleation by mineral dusts

Shown in Fig. 9 and 10 are the fraction frozen curves for droplets containing mineral dust INSs with and without 0.05 M (NH$_4$)$_2$SO$_4$ and fraction frozen curves for ultrapure water (laboratory blanks). The laboratory blanks froze at significantly colder temperatures than the mineral dust samples, indicating that the mineral dust was responsible for ice nucleation in the droplets containing mineral dust.

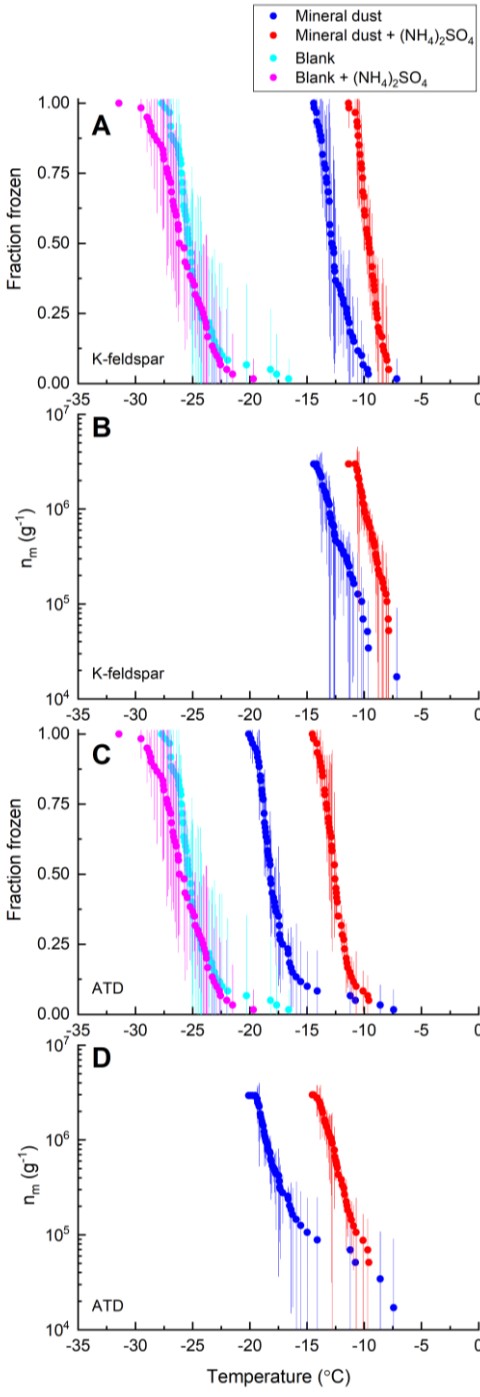

**Figure 9.** Fraction of droplets frozen (A, C) and ice nucleation active site densities $n_m(T)$ (B, D) for samples of Arizona Test Dust (A, B) and K-feldspar (C, D) with (red) and without (blue) 0.05 M (NH4)2SO4 compared with an ultrapure water blank with (pink) and without (cyan) 0.05 M (NH4)2SO4. Freezing data (both samples and blanks) have been corrected for freezing point depression by 0.05 M (NH4)2SO4. Error bars represent the 95% confidence interval of fraction frozen or $n_m(T)$ at temperature T calculated using the Student's T distribution.

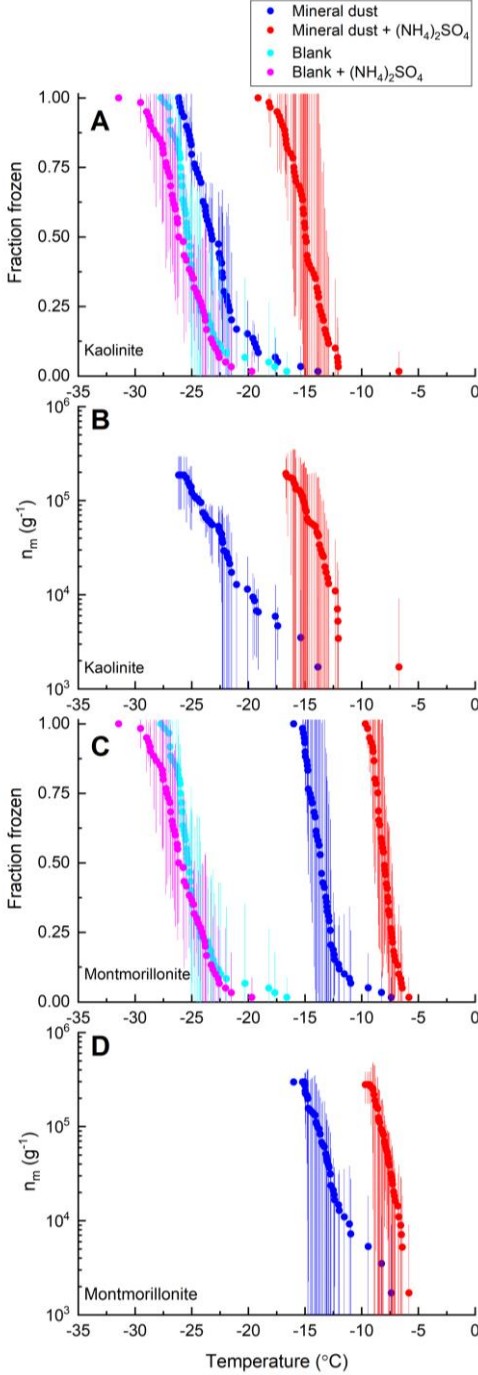

**Figure 10. Fraction of droplets frozen (A, C) and ice nucleation active site densities $n_m(T)$ (B, D) for samples of kaolinite (A, B) and montmorillonite (C, D) with (red) and without (blue) 0.05 M $(NH_4)_2SO_4$ compared with an ultrapure water blank with (pink) and without (cyan) 0.05 M $(NH_4)_2SO_4$. Freezing data (both samples and blanks) have been corrected for freezing point depression by 0.05 M $(NH_4)_2SO_4$. Error bars represent the 95% confidence interval of fraction frozen or $n_m(T)$ at temperature T calculated using the Student's T distribution.**

385

The fraction frozen curves for droplets containing K-rich feldspar dust in the presence of $(NH_4)_2SO_4$ were shifted to warmer
temperatures than those without $(NH_4)_2SO_4$ (Fig. 8, 9). The $\Delta T_{50}$, $\Delta T_{25}$, and $\Delta T_{75}$ values for K-feldspar were 3.21 ± 0.83 °C,
2.73 ± 1.15 °C, and 3.17 ± 0.84 °C, respectively. The fraction frozen curves for droplets containing ATD in the presence of
$(NH_4)_2SO_4$ were also shifted to warmer temperatures than those without $(NH_4)_2SO_4$ (Fig. 8, 9). The $\Delta T_{50}$, $\Delta T_{25}$, and $\Delta T_{75}$
values for ATD were 5.56 ± 0.49 °C, 5.21 ± 1.07 °C, and 5.60 ± 0.82 °C, respectively. The fraction frozen curves for
droplets containing kaolinite in the presence of $(NH_4)_2SO_4$ were also shifted to warmer temperatures than those without
$(NH_4)_2SO_4$ (Fig. 8, 10). The $\Delta T_{50}$, $\Delta T_{25}$, and $\Delta T_{75}$ values for kaolinite were 8.22 ± 3.15 °C, 7.69 ± 2.88 °C, and 9.02 ± 2.30
°C, respectively. Lastly, the fraction frozen curves for droplets containing montmorillonite in the presence of $(NH_4)_2SO_4$
were also shifted to warmer temperatures than those without $(NH_4)_2SO_4$ (Fig. 8, 10). The $\Delta T_{50}$, $\Delta T_{25}$, and $\Delta T_{75}$ values for
montmorillonite were 5.61 ± 1.90 °C, 5.47 ± 1.47 °C, and 5.77 ± 1.61 °C, respectively.

Also shown in Fig. 9 and 10 are $n_m(T)$ curves for droplets containing mineral dust INSs with and without 0.05 M
$(NH_4)_2SO_4$ to facilitate comparison with other studies and between the different mineral INSs used herein. For K-feldspar,
ATD, kaolinite, and montmorillonite, the presence of $(NH_4)_2SO_4$ increased the $n_m(T)$ values by a factor of approximately
10, 20, 30, and 10 respectively.

Several studies have investigated the effects of $(NH_4)_2SO_4$ on ATD, K-feldspar, and kaolinite, the results of which are
summarized in Table S2. These experiments include different concentrations of $(NH_4)_2SO_4$, different concentrations of
mineral dust INSs, and different freezing temperature regimes. We have only included results up to 0.1 M $(NH_4)_2SO_4$ in
Table S2 since this is the range most relevant for our studies. Our results for mineral dust particles are also included in Table
S2 for comparison. Overall, the trends remain quite consistent for most of the minerals. Most notably, a consistent significant
increase in ice nucleation activity is observed for most of these minerals in the presence of $(NH_4)_2SO_4$ (Table S2). ATD is the
only exception, with our study observing $\Delta T_{50}$ value of approximately 5.5 °C while Whale et al., (2018) observed no
significant change in freezing temperature in the presence of $(NH_4)_2SO_4$ (Table S2). Potential reasons for this discrepancy in
results include the following: 1) Differences in the exposure time of ATD particles to $(NH_4)_2SO_4$ before the freezing
experiments were performed (shorter interaction times between $(NH_4)_2SO_4$ and ATD were used in the current study
compared to Whale et. al. 2018), and 2) differences in the composition of the ATD sample used. Although the same type of
ATD was used in both studies, there can be variability in ATD composition from batch to batch (Kaufmann et al., 2016). No
previous studies examined the effects of $(NH_4)_2SO_4$ on montmorillonite. However, the effects of ammonia gas on the ice
nucleation properties of montmorillonite in the deposition mode have been studied and exposure to ammonia gas has been
shown to enhance the ice nucleation activity of montmorillonite (Salam et al., 2007).

All our studies were carried out with $(NH_4)_2SO_4$ concentrations of 0.05 M. Based on previous studies, the freezing ability of mineral dust INSs increase as the $(NH_4)_2SO_4$ concentrations increase up to about 0.1 M, after which, the freezing ability of mineral dust INSs can decrease, presumably by blocking ice nucleation sites at higher concentrations (Whale et al., 2018; Kumar et al., 2018, 2019b). As an example, Kumar et al., (2018) studied the effect of $(NH_4)_2SO_4$ concentrations on the ice

nucleating ability of K-feldspar and observed that as the concentration of $(NH_4)_2SO_4$ increased from ~9x10$^{-6}$ M to ~0.04 M the change in onset freezing temperature upon the addition of $(NH_4)_2SO_4$ ($\Delta T_{onset}$) increased from ~1 to ~4.5 °C. However, as $(NH_4)_2SO_4$ concentration increased from ~0.05 M to ~1 M, $\Delta T_{onset}$ decreased to ~ -12 °C, becoming negative at a concentration of approximately 0.4 M.

### 3.3 Mechanisms

The K-rich feldspar sample studied here consists of 85% microcline ($KAlSi_3O_8$) and 15% albite ($NaAlSi_3O_8$). Since, in general, microcline is a better ice nucleus than albite, microcline is likely responsible for ice nucleation in our experiments (Harrison et al., 2016; Peckhaus et al., 2016b; Zolles et al., 2015; Atkinson et al., 2013; Whale et al., 2017). Microcline is composed of $AlO_4^-$ and $SiO_4$ tetrahedral groups, with the negative charge of $AlO_4^-$ being compensated by $K^+$ in the crystal lattice. When microcline is exposed to $(NH_4)_2SO_4$, ion exchange can occur between $NH_4^+$ and $K^+$ (Gülgönül et al., 2012;

Nash and Marshall, 1957; Stumm and Morgan, 1971; Barker, 1964):

$$KAlSi_3O_8 + NH_4^+ \leftrightarrow NH_4AlSi_3O_8 + K^+ \tag{1}$$

One possible explanation for the strong positive effect of $(NH_4)_2SO_4$ on the ice nucleation ability of K-rich feldspar is ion exchange between $NH_4^+$ in solution and parent $K^+$ in the microcline to form surfaces more conducive to ice nucleation.

In our experiments, the surface of K-rich feldspar was negatively charged since the pH of the solutions were above the point of zero charge (PZC) of K-rich feldspar (~ 1-2) (Kosmulski, 2009). Under these conditions $NH_4^+$ is likely to adsorb to the negatively charged K-rich feldspar surfaces (Kumar et al., 2018; Nash and Marshall, 1957). Therefore, another possible explanation for the strong positive effect of $(NH_4)_2SO_4$ on the ice nucleation ability of K-rich feldspar is the adsorption of $NH_4^+$ on the K-rich feldspar surface, which can potentially affect the orientation of water molecules near the mineral surface

and enhance ice nucleation (Kumar et al., 2018; Anim-Danso et al., 2016).

Kaolinite ($Al_2Si_2O_5(OH)_5$, negatively charged with PZC ~3-6) only offers adsorption of $NH_4^+$ due to lack of charge-balancing cation, while ion exchange and absorption of $NH_4^+$ are both possible on feldspars. Since feldspars and kaolinite constitute a major part of ATD, they may explain the enhanced ice nucleation ability of ATD in the presence of $(NH_4)_2SO_4$.

Although PZC values for montmorillonite can range from 1-10, they are typically in the range of 2-4 which is less than the pH of the solution (Kosmulski, 2009, 2020). Hence $NH_4^+$ is also likely to adsorb to a negatively charged surface of

montmorillonite, which can potentially enhance ice nucleation at the mineral surface, as discussed above. In contrast, these mechanisms do not appear to be important for ice nucleation of the non-mineral dust INSs studied here.

The ice nucleation activity in *P. syringae* and *X. campestris*, the two bacteria studied here, has been successfully attributed to proteins located on the outer cell membrane (Kawahara, 2002; Warren and Corotto, 1989; Wolber and Warren, 1989; Failor et al., 2017; Pummer et al., 2015; Zhao and Orser, 1990). The proteins have repeating units containing threonine amino acids that provide a template for ice (Kawahara, 2002; Warren and Corotto, 1989; Wolber and Warren, 1989; Gurian-Sherman and Lindow, 1993; Hew and Yang, 1992; Zhao and Orser, 1990; Graether and Jia, 2001; Garnham et al., 2011). Either

$(NH_4)_2SO_4$ molecules in solution do not interact with the repeating unit of the protein, or any interactions that occur (e.g., adsorption of ions or the induction of small conformational changes) do not lead to a significant enhancement of the ice nucleation activity of these proteins.

In comparison to *P. syringae* and *X. campestris*, the molecules responsible for ice nucleation in the other non-mineral dust

samples are not well studied and their precise identities and mechanisms for ice nucleation are not well understood. Regardless of the specific molecules an mechanisms responsible for the ice nucleation activity of the studied non-mineral dust INSs, our results show that either the non-mineral dust INSs studied do not interact with the $(NH_4)_2SO_4$ molecules in solution, or the mechanisms of ice nucleation by these INSs are such that any interactions that may occur (e.g. adsorption of ions or the induction of small conformational changes) do not have a significant impact, since changes observed in ice

nucleation activity of the large majority of the non-mineral dust INSs were less than the uncertainty of our measurements.

**4 Conclusions**

In this work, immersion freezing experiments were performed using the droplet freezing technique to investigate the effects of a low concentration (0.05 M) of $(NH_4)_2SO_4$, an atmospherically relevant inorganic solute, on the ice nucleating ability of a range of non-mineral dust INSs of atmospheric relevance. These included bacteria, fungi, exudates from sea ice diatoms

collected in Antarctica, INSs from the sea surface microlayer collected in the Arctic, and humic and fulvic acid. For the majority of these non-mineral dust INSs, $\Delta T_{50}$, $\Delta T_{25}$, and $\Delta T_{75}$ values were less than the uncertainty in the measurements, indicating no change in freezing temperature in the presence of $(NH_4)_2SO_4$. Only two samples (*X. campestris* and Antarctic diatom exudates) showed significantly negative values of $\Delta T_{50}$ and/or $\Delta T_{75}$. In no case was a statistically significant positive $\Delta T_{50}$, $\Delta T_{25}$, or $\Delta T_{75}$ observed for a non-mineral dust INS, consistent with $(NH_4)_2SO_4$ either having no effect on or very

slightly decreasing the ice nucleation ability of non-mineral dust INSs. As a comparison, we also investigated the effects of $(NH_4)_2SO_4$ on the ice nucleating ability of four types of mineral dust (ATD, K-rich feldspar, montmorillonite, and kaolinite). All mineral dusts had significantly positive values for $\Delta T_{50}$, $\Delta T_{25}$, and $\Delta T_{75}$ between 3 °C and 9 °C and showed an increase in $n_m(T)$ by a factor of ~10 to ~30 indicating that the addition of $(NH_4)_2SO_4$ enhances the ice nucleation activity of these

mineral dust species. The different responses of mineral dust and non-mineral dust INSs to the addition of $(NH_4)_2SO_4$

suggest that they have different mechanisms of ice nucleation and/or different interactions with the solute at the ice nucleating surface. This is supported by the existing literature on the mechanisms of ice nucleation by mineral dusts, particularly K-rich feldspar, and proteins found in the studied bacteria *P. syringae* and *X. campestris*.

Our results suggest that the freezing temperature response of samples of unknown composition to the addition of 0.05 M

$(NH_4)_2SO_4$ could potentially be used as an assay for the presence of mineral dust INSs in atmospheric samples. At this $(NH_4)_2SO_4$ concentration the ice nucleating ability of several types of mineral dusts were enhanced, but the ice nucleating ability of several atmospherically relevant non-mineral dust INSs were not affected. Additional measurements are needed with a range of mineral dust and non-mineral dust INS concentrations to confirm that this assay is appropriate for a range of mineral dust and non-mineral dust INS concentrations. Nevertheless, a comparison with results in the literature does suggest

that our results may be applicable to a range of mineral dust and non-mineral dust INS concentrations.

In addition, our results suggest that the relative importance of mineral dust to non-mineral dust INSs for ice nucleation in mixed-phase clouds could increase as these particles become coated with $(NH_4)_2SO_4$ in the atmosphere. Furthermore, our results provide additional evidence that the ice nucleating ability of mineral dust INSs in the immersion freezing mode

increases when coated with $(NH_4)_2SO_4$, and hence this process should be included in models used to predict concentrations of INSs in mixed-phase clouds. The concentrations of $(NH_4)_2SO_4$ in mixed-phase clouds can be greater or less than 0.05 M. As a result, for atmospheric predictions, additional studies at concentrations of $(NH_4)_2SO_4$ less than and greater than 0.05 M are needed. For atmospheric predictions, additional studies with a range of mineral dust and non-mineral dust INS concentrations are also needed. Our study represents a valuable initial survey on the effect of $(NH_4)_2SO_4$ on ice nucleation by

a wide range of INS types, many of which have not been previously studied.

*Author contribution*: SEW, JY, and AKB planned the experiments. SW conducted the experiments and prepared the initial paper draft. CX assisted with the experiments. PA provided the *X. campestris* strain. VEI collected the SML samples. JC prepared all bacterial and fungal cultures. SEW, AK, YX, JY, and AKB contributed to the interpretation of the results. SEW,

AK, YX, and AKB wrote the manuscript. All co-authors reviewed and provided comments on the paper.

*Competing interests*: The authors declare that they have no conflict of interest

*Acknowledgements*: The authors thank Steven Lindow for providing *P. syringae* Strain 31R1. The authors thank James

Raymond for collecting and providing the Antarctic sea ice diatom exudate sample.

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
