# Peer review of "The effect of $(\text{NH}_4)_2\text{SO}_4$ on the freezing properties of non-mineral dust ice nucleating substances of atmospheric relevance"

_Atmospheric Chemistry and Physics, 2021_

## Author Comment (AC1)

Prof. Thomas Koop

Executive Editor of Atmospheric Chemistry and Physics

Dear Thomas,

Listed below are our responses to the comments provided by the reviewers of our manuscript. For clarity and visual distinction, the comments or questions from reviewers are in black text, and are preceded by bracketed, italicized numbers (e.g. *[1]*). Our (authors') responses are in red text below each comment or question with matching italicized numbers (e.g. *[A1]*). We thank the reviewers for carefully reading our manuscript and for their helpful comments!

Sincerely,

Allan Bertram

Professor of Chemistry

University of British Columbia

**Reviewer: 1**

Comments:

The manuscript presented the effect of ammonium sulfate (AS) on ice nucleation ability of a set of non-mineral dust substances via immersion freezing. This study investigated the immersion freezing of bacteria, fungi, sea ice diatom exudates, sea surface microlayer, and humic substances in dilute AS solution. For comparison, the effect of AS on immersion freezing of four types of mineral dust were also investigated. The manuscript showed that there is no significant change for most of the tested non-mineral dust substances, expect for bacteria X. Campestris; for the tested mineral dusts, there is an increase in the median freezing temperatures, ranging from 3 K to 8 K. This study provides additional data sets for the better understanding in the ice nucleation potential of different types of substances and the effect of additional AS. The manuscript is well written and suitable for the publication in this journal. A few issues and comments need to be considered before publication.

*[1]*, Line 170-200, for mineral dusts, the weight percentages of dust in the solution used here are not the same, can the authors comment on why different concentrations were used for preparing the droplets? What are the potential effects of AS on the freezing temperature of droplets with different concentrations of dust?

*[A1]* The weight percentage used in the mineral dust experiments was adjusted to ensure that freezing occurred at temperatures warmer than the blanks (> 20 °C). To achieve this goal, the following concentrations were used for kaolinite, montmorillonite, feldspar, and ATD: 1 wt %, 1 wt %, 0.1 wt %, and 0.1 wt %, respectively. This information will be added to the revised manuscript. We will also normalize the freezing data to the mass of the mineral dust used in the experiments in the revised manuscript. By normalizing the freezing data to the mass, the effect of the different concentrations of dust on the freezing ability is accounted for. Also, in the revised manuscript we will discuss the effect of mineral dust concentrations on the results.

*[2]* In Figure 2 to Figure 5, the frozen fraction for the Blank and Blank+AS data presented in different figures are somehow different. Were these blank experiments done at different time periods? In Line 228-231, it was mentioned that only the heterogeneous freezing temperatures have been corrected, have the blank data also been corrected?

*[A2]* In Figures 2 to 5, some of the blanks correspond to growth media controls, while the other blanks correspond to ultrapure water controls. Both the blanks and INS suspensions were corrected for freezing point depression by ammonium sulfate. In the revised manuscript we will modify the text and figure captions to make these points clear.

*[3]* In Section 3.2, as mentioned in several places that the results presented in this study are consistent with previous studies, e.g., Line 300, 316, 322. I would suggest to summarize these data and present in a table. This table or summary may further support the claim that freezing temperature response of unknown substances to additional AS could be used as a "fingerprint" for the presence of mineral dust.

*[A3]* As suggested, in the revised manuscript we will add a new table that summarizes previous studies that have investigated the effect of ammonium sulfate on the freezing properties of mineral dust. We will also add an additional table that summarizes previous studies that investigated the effect of ammonium sulfate on the freezing properties of non-mineral dust INSs.

*[4]* For Figure 2-5 and Figure 7, for comparison purpose, it is great to show the data for different trials and see the variations, but the author may need to consider summarizing the data from different trials, I think that is the final form of data sets which readers may use.

*[A4]* To address the referee's comments, in the revised manuscript we will plot freezing data as averages with 95 % confidence intervals from the different trials.

**Reviewer 2:**

Comments:

This paper presents results from experiments investigating the impact of ammonium sulfate additions on the freezing properties observed for suspensions containing dust and non-dust INPs. The paper has a clear structure and is well written. The visual presentation of experimental results is adequate. The findings presented in this study are interesting, but a much larger emphasis should be put on enabling the translation of these findings into atmospheric implications, i.e. through a more thorough analysis delivering a quantification of the measured effects which is not dependent on the experimental parameters chosen for this study (e.g. dust mass).

The following points need to be addressed before this paper can be published:

*[5]* To be able to compare against other experimental studies more easily, results should also be reported as ice nucleation active surface site densities (i.e. normalized to mass or surface), so that at least relative changes at a fixed ammonium sulfate concentration could be quantified independently from the INP concentrations. Ideally, I would like to see measurements at different INP and ammonium sulfate concentrations, so that these results could be used more readily in climate models.

*[A5]* To address the referee's comment, in the revised manuscript, in most cases (mineral dust, bacteria, fungi, humic substances) we will present our results in terms of ice nucleation active site densities per gram of material $n_m$ (i.e., we will normalize the freezing results to mass).  For a few of the cases (Antarctic diatom exudates and SML samples) the mass of the ice nucleating material is not known, so we will report the concentrations of INS per liter of solution $[INS(T)]$ (i.e., we will normalize the freezing results to volume of liquid).  To address the referee's comment regarding measurements at different INS and ammonium sulfate concentrations, in the revised manuscript we will discuss the effect of ammonium sulfate concentrations and INS concentrations on the trends observed in the current study. We will also point out

that for atmospheric predictions, additional studies as a function of ammonium sulfate concentrations and INS concentrations are needed.

*[6]* Also, there is no discussion of statistical uncertainties associated with the observed frozen fractions.

*[A6]* To address the referee's comments, in the revised manuscript we will report the average fraction frozen results with 95% confidence intervals rather than the results from each trial.

*[7]* Lastly, I don't agree with the authors' conclusion that ammonium sulfate solutions can be used to reliably detect the presence of dust INPs. There are no experimental results showing how sensitive this method would be to variations in concentration (INPs, ammonium sulfate). Along the same lines, it is not clear what the results would be for complex particles, e.g. agricultural soil dusts that are mixtures of mineral dust and organics.

*[A7]* The ammonium sulfate assay we are suggesting would only use a concentration of 0.05 M ammonium sulfate to detect the presence of mineral dust INSs in atmospheric samples. Hence, an understanding of the sensitivity to variations in ammonium sulfate concentrations is not needed for the suggested ammonium sulfate assay. In the revised manuscript we will make this clear. We agree that additional measurements are needed with a range of mineral dust and non-mineral dust INS concentrations to confirm that an assay based on the response to 0.05 M ammonium sulfate can reliably be used to detect the presence of mineral dust INPs. This point will also be made clear in the revised manuscript.

**Specific comments:**
*[8]* l. 90: Reference for selected ammonium sulfate concentration?

*[A8]* The following reference will be added to the revised manuscript at this location:
Reischel, M. T. and Vali, G.: Freezing nucleation in aqueous electrolytes, 27, 414–427, https://doi.org/10.1111/j.2153-3490.1975.tb01692.x, 1975.

*[9]* Fig. 2-5: How do you explain the substantial variability in the blank measurements or between different microlayer samples? And how does this variability impact the interpretation of your experimental results?

*[A9]* There was some variability observed within the ultrapure water laboratory blanks at low fraction frozen values, which we believe to be a result of possible defects or imperfections in the surfaces of our

hydrophobic glass slides or possible impurities present in the ultrapure water. For the microlayer samples, we expect significant variability between Station 2 and Station 7 because they are from two very different geographical locations. The variability between different trials of one microlayer sample (e.g. SML station 2) may be due to a lack of homogeneity of the samples, although we cannot confirm. Nevertheless, this variability should not impact the interpretation in our results since the freezing of the samples occurred at temperatures warmer than the blanks and the variability in the freezing results were considered in the error analysis.

**Technical comment:**

[10] Fig. 5: Please change 'control' to 'blank'.

*[A10]* This change will be made in the revised manuscript.

---

## Author Response (AR1)

Prof. Thomas Koop

Executive Editor of Atmospheric Chemistry and Physics

Dear Thomas,

Listed below are our responses to the comments provided by the reviewers of our manuscript. For clarity and visual distinction, the comments or questions from reviewers are in black text, and are preceded by bracketed, italicized numbers (e.g. *[1]*). Our (authors') responses are in red text below each comment or question with matching italicized numbers (e.g. *[A1]*). We thank the reviewers for carefully reading our manuscript and for their helpful comments!

Sincerely,

Allan Bertram

Professor of Chemistry

University of British Columbia

**Reviewer: 1**

Comments:

The manuscript presented the effect of ammonium sulfate (AS) on ice nucleation ability of a set of non-mineral dust substances via immersion freezing. This study investigated the immersion freezing of bacteria, fungi, sea ice diatom exudates, sea surface microlayer, and humic substances in dilute AS solution. For comparison, the effect of AS on immersion freezing of four types of mineral dust were also investigated. The manuscript showed that there is no significant change for most of the tested non-mineral dust substances, expect for bacteria X. Campestris; for the tested mineral dusts, there is an increase in the median freezing temperatures, ranging from 3 K to 8 K. This study provides additional data sets for the better understanding in the ice nucleation potential of different types of substances and the effect of additional AS. The manuscript is well written and suitable for the publication in this journal. A few issues and comments need to be considered before publication.

*[1]*, Line 170-200, for mineral dusts, the weight percentages of dust in the solution used here are not the same, can the authors comment on why different concentrations were used for preparing the droplets? What are the potential effects of AS on the freezing temperature of droplets with different concentrations of dust?

*[A1]* The weight percentage used in the mineral dust experiments was adjusted to ensure that freezing occurred at temperatures warmer than the blanks (> 20 °C). To achieve this goal, the following concentrations were used for kaolinite, montmorillonite, feldspar, and ATD: 1 wt %, 1 wt %, 0.1 wt %, and 0.1 wt %, respectively. This information has been added to the revised manuscript. We have also normalized the freezing data to the mass of the mineral dust used in the experiments in the revised manuscript. By normalizing the freezing data to the mass, the effect of the different concentrations of dust on the freezing ability is accounted for. Also, in the revised manuscript we have discussed the effect of mineral dust concentrations on the results. The following are specific text added to the revised manuscript:

"For the mineral dust samples, concentrations were adjusted to ensure that freezing occurred at temperatures warmer than the blanks (> 20 °C). As a result, some mineral dust suspensions (kaolinite and montmorillonite suspensions) were prepared at different concentrations than used for K-rich feldspar (see below). For all the mineral dust samples, we used ultrapure water as the blanks."

"Several studies have investigated the effects of $(NH_4)_2SO_4$ on ATD, K-feldspar, and kaolinite, the results of which are summarized in Table S2. These experiments include different concentrations of $(NH_4)_2SO_4$, different concentrations of mineral dust INSs, and different freezing temperature regimes. … Overall, the trends remain quite consistent for most of the minerals. Most notably, a consistent significant increase in ice nucleation activity is observed for most of these minerals in the presence of $(NH_4)_2SO_4$ (Table S2)."

"Additional measurements are needed with a range of mineral dust and non-mineral dust INS concentrations to confirm that this assay is appropriate for a range of mineral dust and non-mineral dust INS concentrations. Nevertheless, a comparison with results in the literature does suggest that our results may be applicable to a range of mineral dust and non-mineral dust INS concentrations."

*[2]* In Figure 2 to Figure 5, the frozen fraction for the Blank and Blank+AS data presented in different figures are somehow different. Were these blank experiments done at different time periods? In Line 228-231, it was mentioned that only the heterogeneous freezing temperatures have been corrected, have the blank data also been corrected?

*[A2]* In Figures 2 to 5, some of the blanks correspond to growth media controls, while the other blanks correspond to ultrapure water controls. Both the blanks and INS suspensions were corrected for freezing point depression by ammonium sulfate. In the revised manuscript we have modified the text and figure captions to make these points clear. Please see revised figure captions for details.

*[3]* In Section 3.2, as mentioned in several places that the results presented in this study are consistent with previous studies, e.g., Line 300, 316, 322. I would suggest to summarize these data and present in a table. This table or summary may further support the claim that freezing temperature response of unknown substances to additional AS could be used as a "fingerprint" for the presence of mineral dust.

*[A3]* As suggested, in the revised manuscript we have added a new table that summarizes previous studies that have investigated the effect of ammonium sulfate on the freezing properties of mineral dust (Table S2). We have also added an additional table that summarizes previous studies that investigated the effect of ammonium sulfate on the freezing properties of non-mineral dust INSs (Table S1).

*[4]* For Figure 2-5 and Figure 7, for comparison purpose, it is great to show the data for different trials and see the variations, but the author may need to consider summarizing the data from different trials, I think that is the final form of data sets which readers may use.

*[A4]* To address the referee's comments, in the revised manuscript we have plotted freezing data as averages with 95 % confidence intervals from the different trials (Figures 2-7, 9-10).

**Reviewer 2:**

Comments:

This paper presents results from experiments investigating the impact of ammonium sulfate additions on the freezing properties observed for suspensions containing dust and non-dust INPs. The paper has a clear structure and is well written. The visual presentation of experimental results is adequate. The findings presented in this study are interesting, but a much larger emphasis should be put on enabling the translation of these findings into atmospheric implications, i.e. through a more thorough analysis delivering a

quantification of the measured effects which is not dependent on the experimental parameters chosen for this study (e.g. dust mass).

The following points need to be addressed before this paper can be published:

*[5]* To be able to compare against other experimental studies more easily, results should also be reported as ice nucleation active surface site densities (i.e. normalized to mass or surface), so that at least relative changes at a fixed ammonium sulfate concentration could be quantified independently from the INP concentrations. Ideally, I would like to see measurements at different INP and ammonium sulfate concentrations, so that these results could be used more readily in climate models.

*[A5]* To address the referee's comment, in the revised manuscript, in most cases (mineral dust, bacteria, fungi, humic substances) we have presented our results in terms of ice nucleation active site densities per gram of material $n_m$ (i.e., we have normalized the freezing results to mass) (Figures 2-4, 7, 9-10). For a few of the cases (Antarctic diatom exudates and SML samples) the mass of the ice nucleating material is not known, so we have reported the concentrations of INS per liter of solution $[INS(T)]$ (i.e., we have normalized the freezing results to volume of liquid) (Figures 5-6). To address the referee's comment regarding measurements at different INS and ammonium sulfate concentrations, in the revised manuscript we have discussed the effect of ammonium sulfate concentrations and INS concentrations on the trends observed in the current study. We have also pointed out that for atmospheric predictions, additional studies as a function of ammonium sulfate concentrations and INS concentrations are needed. The following are specific text added to the revised manuscript:

"A few previous studies have also investigated the effect of $(NH_4)_2SO_4$ on non-mineral INSs (specifically Snomax, humic acid and leaf derived nuclei). The results from these previous studies are summarized in Table S1. We have only included results up to 0.1 M $(NH_4)_2SO_4$, since this is the range most relevant for our experiments. For comparison purposes we have also summarized our results for Snomax and humic acid in Table S1. The previous experiments include measurements with different concentrations of $(NH_4)_2SO_4$, different concentrations of non-mineral INSs, and different freezing temperature regimes. In no case was ice nucleation enhanced by the addition of $(NH_4)_2SO_4$, consistent with our results.

[revised manuscript text omitted]

*[6]* Also, there is no discussion of statistical uncertainties associated with the observed frozen fractions.

*[A6]* To address the referee's comments, in the revised manuscript we have reported the average fraction frozen results with 95% confidence intervals rather than the results from each trial (Figures 2-7, 9-10).

*[7]* Lastly, I don't agree with the authors' conclusion that ammonium sulfate solutions can be used to reliably detect the presence of dust INPs. There are no experimental results showing how sensitive this method would be to variations in concentration (INPs, ammonium sulfate). Along the same lines, it is not clear what the results would be for complex particles, e.g. agricultural soil dusts that are mixtures of mineral dust and organics.

*[A7]* The ammonium sulfate assay we are suggesting would only use a concentration of 0.05 M ammonium sulfate to detect the presence of mineral dust INSs in atmospheric samples. Hence, an understanding of the sensitivity to variations in ammonium sulfate concentrations is not needed for the suggested ammonium sulfate assay. We have made this clear in the revised manuscript. We agree that additional measurements are needed with a range of mineral dust and non-mineral dust INS concentrations to confirm that an assay based on the response to 0.05 M ammonium sulfate can reliably be used to detect the presence of mineral dust INPs. This point has also been made clear in the revised manuscript. The following are specific text added to the revised manuscript:

"This difference also suggests that the addition of $(NH_4)_2SO_4$ (0.5 M) to atmospheric samples of unknown composition could potentially be used as an indicator or assay for the presence of mineral dust ice nuclei,

although additional studies are still needed as a function of concentration of ice nucleating substance (INS) to confirm the same trends are observed for different INS concentrations than used here. A comparison with results in the literature does suggest that our results may be applicable to a range of mineral dust and non-mineral dust INS concentrations."

"Our results suggest that the freezing temperature response of samples of unknown composition to the addition of 0.05 M $(NH_4)_2SO_4$ could potentially be used as an assay for the presence of mineral dust INSs in atmospheric samples. At this $(NH_4)_2SO_4$ concentration the ice nucleating ability of several types of mineral dusts were enhanced, but the ice nucleating ability of several atmospherically relevant non-mineral dust INSs were not affected. Additional measurements are needed with a range of mineral dust and non-mineral dust INS concentrations to confirm that this assay is appropriate for a range of mineral dust and non-mineral dust INS concentrations. Nevertheless, a comparison with results in the literature does suggest that our results may be applicable to a range of mineral dust and non-mineral dust INS concentrations."

**Specific comments:**

*[8]* l. 90: Reference for selected ammonium sulfate concentration?

*[A8]* The following reference has been added to the revised manuscript at this location:

Reischel, M. T. and Vali, G.: Freezing nucleation in aqueous electrolytes, 27, 414–427, https://doi.org/10.1111/j.2153-3490.1975.tb01692.x, 1975.

*[9]* Fig. 2-5: How do you explain the substantial variability in the blank measurements or between different microlayer samples? And how does this variability impact the interpretation of your experimental results?

*[A9]* There was some variability observed within the blanks at low fraction frozen values, which we believe to be a result of possible defects or imperfections in the surfaces of our hydrophobic glass slides or possible impurities present in the ultrapure water or culture blanks. For the microlayer samples, we expect significant variability between Station 2 and Station 7 because they are from two very different geographical locations. The variability between different trials of one microlayer sample (e.g. SML station 2) may be due to a lack of homogeneity of the samples, although we cannot confirm. Nevertheless, this variability should not impact the interpretation in our results since the freezing of the samples occurred at temperatures warmer than the blanks and the variability in the freezing results were considered in the error analysis.

**Technical comment:**

[10] Fig. 5: Please change 'control' to 'blank'.

*[A10]* This change has been made in the revised manuscript.

---

## Author Response (AR2)

Prof. Thomas Koop
Executive Editor
Atmospheric Chemistry and Physics

Dear Thomas,

We have made both of the changes suggested by the referee.  We thank you and the referee for the helpful feedback!

Sincerely,

Allan Bertram
Professor of Chemistry
University of British Columbia